microbiology/evolution

anellovirus, TTV, group 6,
phylogenetic analysis, Taiwan

**Author for correspondence:**
Hsin-Fu Liu
e-mail: hsinfu@mmh.org.tw

# Detection and genetic characterization of the novel torque teno virus group 6 in Taiwanese general population

Kuang-Liang Hsiao[1,2], Li-Yu Wang[3], Ju-Chien Cheng[5], Yu-Jung Cheng[2], Chiung-Ling Lin[1] and Hsin-Fu Liu[1,2,4]

[1]Department of Medical Research, MacKay Memorial Hospital, Taipei, Taiwan
[2]Department of Bioscience and Biotechnology, National Taiwan Ocean University, Keelung, Taiwan
[3]Department of Medicine, and [4]Institute of Biomedical Sciences, MacKay Medical College, New Taipei City, Taiwan
[5]Department of Medical Laboratory Science and Biotechnology, China Medical University, Taichung, Taiwan

K-LH, 0000-0003-4397-7762; H-FL, 0000-0003-0082-2269

Torque teno virus (TTV) is one of the most common human viruses and can infect an individual with multiple genotypes chronically and persistently. TTV group 6 is a recently discovered phylogenetic group first isolated from eastern Taiwan indigenes, but whether the TTV group 6 was also prevalent in the general population still unknown. One hundred and three randomly collected blood samples from general population and 66 TTV positive DNA samples extracted from Taiwan indigenes were included. A group-6-specific PCR was developed for re-screen over TTV positive samples. Two TTV group 6 positive samples from general population were cloned and sequenced for identifying mix-infected TTVs and confirming their classification by maximum-likelihood and Bayesian inference phylogeny. TTV group 6 can be detected in 4.5% (4/89) and 7.6% (5/66) of TTV positive samples from Taiwanese general population and eastern Taiwan indigenes, respectively. Sample VC09 was mix-infected with TTV groups 3 and 6. Sample VC99 was mix-infected with TTV groups 3, 4 and 6. A highly diverse triple overlapping region was observed, which may represent a unique phenomenon of TTV. The group-6-specific PCR can successfully detect TTV group 6. TTV group 6 may be prevalent worldwide regardless of the geographic region and/or ethnic groups.

## 1. Introduction

Torque teno virus (TTV) is the first circular negative-stranded DNA virus that has been found to infect humans with multiple

genotypes, chronically, persistently, ubiquitously, and can be detected from early in life [1–3]. TTV is considered as an 'orphan' virus within the family *Anelloviridae* and also one of the most common and the most abundant viruses in human blood [4,5]. Although it has not been confirmed to be associated with any disease so far, it is worthwhile to develop group-specific detection methods.

TTV genome is around 3800 nucleotides (nt) in length, which include a large open read frame (ORF): ORF1, at least two other overlapped open reading frames, and a GC-rich untranslated region [2,6–8]. The ORF1 possesses about two-thirds of whole viral genome in length and has a very high diversity despite being a DNA virus. TTV can be divided into numerous species and seven phylogenetic groups by phylogenetic analysis based on this region [9].

TTV group 1, included the prototype TA278, was first isolated in Japan. Soon after, TTV group 1 was also identified in West Africa, North America, East Asia and Europe [1,10–16]. TTV group 2 was majorly isolated in Europe but also in Central Asia and Indonesia [12,17–21]. TTV group 3 can be found in America, East Asia and Europe [12,15,16,22–26]. TTV group 4 has relatively few sequences currently, but is also distributed across Asia and Europe, in China, Japan and Germany [23,26,27]. TTV group 5 was first found in East Asia and subsequently identified in North America and Europe [12,16,27–29]. Since TTV groups' geographic distribution listed here was obviously biased by research's sampling location, it is generally considered that TTVs was distributed globally without geographic specificity [30].

The recently proposed phylogenetic groups: group 7 was found at least in both Taiwan and China, whereas group 6 was only identified in eastern Taiwan indigenes [9]. Owing to the geographic distribution of other TTV groups, it is highly possible that group 6 is not only in Taiwanese indigenes but also should be in the Taiwanese general population. To clarify this issue, we developed a group-6-specific PCR for re-screen over TTV positive samples. The mix-infection manner within TTV group 6 positive samples and a detailed genetic characterization of these sequences were also studied.

# 2. Methods

## 2.1. Serum samples and DNA extraction

One hundred and three anonymous surplus blood samples were randomly collected from Health Examination Center, MacKay Memorial Hospital Taipei, Taiwan, and serum was separated from the whole blood samples. Viral DNA was extracted from 100 µl of serum by High Pure Viral Nucleic Acid Kit (Roche). The serum samples were adjusted to a volume of 200 µl with phosphate-buffered saline to fit the manufacturer's instructions. Extracted viral DNAs were used as templates for PCR reactions without measuring their concentration, because these DNA extracts were expected as a mixture of various viral DNAs. In addition, a total of 66 remaining TTV positive DNA extracts from our previous study was also included [9].

## 2.2. Detection of torque teno virus DNA

Overall TTV DNA detection was performed by following the previous study [9] and a random mixture of viral DNA extraction was used as a positive control since the TTV has a prevalence of over 90% among human beings [30].

## 2.3. Detection of torque teno virus group 6 DNA

TTV group-6-specific primers: sense primer 5'-GGA GRA AGA GGA GRG CGG TMG G-3' and antisense primer 5'-TCG TGT CCC CCG AAG TTS GG-3' were deduced from a group-6-specific untranslated region that located near the upstream of ORFs to detect TTV group 6 in studied populations. PCR was performed with AmpliTaq Gold DNA Polymerase (Applied Biosystems) with the final concentration of PCR reaction mix of 1 x PCR Gold Buffer, 1 mM MgCl, 200 µM dNTP, 0.4 µM of each primer and 0.04 unit AmpliTaq GOLD. A 25 µl reaction was set up containing 2.5 µl of viral DNA template, which was the same as the overall TTV DNA detection. The serum viral DNA extract of sample: TW53 was used as positive control which is one of the remaining DNA extracts from our previous study [9]. The amplification conditions were 12 min preheating at 94°C, followed by 40 regular PCR cycles of 30 s at 94°C for denaturing, 30 s at 58°C for annealing and 45 s at 72°C for elongation, and a final extension step of 7 min at 72°C in GeneAmp PCR System 9700 (Applied

Biosystems). The amplicon is according to nucleotide position 60–201 nt of group 6 strain TW53A26 and PCR products were verified by agarose gel electrophoresis and Sanger sequencing.

## 2.4. Amplification and sequencing of torque teno virus ORF1

For confirming the existence of TTV group 6 and verifying their mix-infection state, two of TTV group 6 positive samples were picked randomly for entire ORF1 cloning. The primers and amplicon size for ORF1 amplifying were according to the previous study [9] with a simplified amplification condition. PCR was performed with TaKaRa LA taq (TaKaRa) following the manufacturer's instructions with a final concentration of 0.4 µM of each primer in the PCR reaction mix. A 25 µl reaction was set up containing 2.5 µl of viral DNA template. The amplification conditions of the long-length PCR was started with a preheating of 1 min at 94°C; followed by 35 PCR cycles: 10 s at 98°C for denaturing, 30 s at 63°C for annealing and 2 min 30 s at 72°C for elongation; and followed by a final extension step of 5 min at 72°C in GeneAmp PCR System 9700 (Applied Biosystems). The amplicon is according to nucleotide position 1–2311 nt of group 6 strain TW53A26 and PCR products were visually confirmed by agarose gel electrophoresis. The amplicon was purified from the sliced gel band by using GFX PCR DNA and Gel Band Purification Kit (Amersham Biosciences). Purified PCR products were then inserted into pCR-TOPO 2.1 vector (Invitrogen) and transformed into DH5$\alpha$ competent cells. Plasmid DNA was extracted from each single colony by alkaline lysis and sequenced by primer walking approach. ORF map was drawn by ApE software and putative splicing sites were predicted by ASSP for further analysis [31,32].

## 2.5. Phylogenetic analysis

To determine the phylogenetic relationship of Taiwanese TTVs, a total of 193 TTV ORF1 sequences from GenBank and 14 sequences from this study (electronic supplementary material, table S1) were included for phylogenetic analysis. Nucleotide sequences were aligned by using the MEGA 5 software package on the basis of the ORF1 amino acid sequences to prevent the introduction of nonsense indels [33]. The aligned TTV ORF1 nucleotide sequences were further processed by Gblocks to eliminate ambiguous aligned regions with 'allow all gaps' setting [34]. The concatenated multiplex sequence alignment was automatically generated by Gblocks after processing. Gblocks-processed alignment was used for evolutionary model selection and subjected phylogenetic tree reconstruction.

The best-fit evolutionary model was selected by the Bayesian information criterion, and maximum-likelihood (ML) tree was reconstructed with 1000 bootstrap replications using MEGA 5 software package [33]. Bayesian inference of phylogeny was carried out by MrBayes software using Metropolis-coupled Markov chain Monte Carlo approach [35]. Metropolis-coupled Markov chain Monte Carlo was performed with two parallel runs, each run with four incrementally heated (temperature = 0.1) chains for 3 000 000 generations, and sampled per 500 generations until the standard deviation of the split frequencies falls below 0.01. Bayesian trees were inferred from combined data of both runs with 10% burn-in by maximum clade credibility using TreeAnnotator v. 1.8.4 [36].

## 2.6. Non-synonymous and synonymous substitution analysis

We counted non-synonymous to synonymous substitution (dN/dS) ratios from pairwise coding sequences by PAL2NAL with default parameters [37]. We focused on using dN/dS as a measurement of diversity that reflecting more on changes in the protein sequence.

# 3. Results

## 3.1. Detection of torque teno virus DNA

TTV DNA can be detected in 86.4% (89/103) samples from Taiwanese general population. Among them, 4.5% (4/89) of TTV positive samples were detected as TTV group 6 positive. The estimated prevalence of overall TTV was lower than eastern Taiwan indigenes (95% [133/140] of previous study [9], $p = 0.0185$). The TTV group 6 can be detected in 7.6% (5/66) of remaining TTV positive DNA extraction from previous study, which is slightly higher than general population but not significant.

**Table 1.** Genetic diversity and dN/dS ratio of the TTV clone sequences and their closest sequence from GenBank database. a.a.: amino acid.

| sequences (ORF1 in a.a.) | closest sequence (ORF1 in a.a.) | DNA identity (%) | dN/dS | group (border of intra-group DNA identity[c]) |
|---|---|---|---|---|
| VC99B03 (735 a.a.) | KC197 (734 a.a.) | 81.0 | 0.1630 | 4 (>61.48%) |
| VC99B15 (735 a.a.) | KC197 (734 a.a.) | 81.0 | 0.1645 | 4 (>61.48%) |
| VC09N27 (736 a.a.) | TW53A31 (736 a.a.) | 94.2 | 0.2647 | 6 (>75.96%) |
| VC09N39 (736 a.a.) | TW53A31 (736 a.a.) | 93.8 | 0.2411 | 6 (>75.96%) |
| VC99A01 (713[a] a.a.) | TW53A31 (736 a.a.) | 94.2[b] | 0.2362 | 6 (>75.96%) |
| VC99B09 (736 a.a.) | TW53A31 (736 a.a.) | 94.0 | 0.2702 | 6 (>75.96%) |
| VC99B13 (736 a.a.) | TW53A31 (736 a.a.) | 93.9 | 0.2610 | 6 (>75.96%) |
| VC99B18 (736 a.a.) | TW53A31 (736 a.a.) | 94.0 | 0.2900 | 6 (>75.96%) |
| VC99B21 (736 a.a.) | TW53A31 (736 a.a.) | 94.2 | 0.2515 | 6 (>75.96%) |
| VC99B24 (736 a.a.) | TW53A31 (736 a.a.) | 94.2 | 0.2341 | 6 (>75.96%) |
| VC99B11 (739 a.a.) | LTT10 (740 a.a.) | 88.5 | 0.1052 | 3a (>62.88%) |
| VC09N93 (754 a.a.) | SAa-38 (756 a.a.) | 87.6 | 0.1723 | 3c (>61.18%) |
| VC09N02 (772 a.a.) | P19-2 (772 a.a.) | 86.6 | 0.2043 | 3b (>61.18%) |
| VC09N03 (261[a] a.a.) | P15-1 (775 a.a.) | 89.3[b] | 0.2085 | 3b (>61.18%) |

[a]Premature stops at a.a. 713 and 261 for VC99A01 and VC09N03, respectively, but the downstream region remains conserved.
[b]Pairwise sequence identities were compared with full-length ORF1 of TW53A31 and P15-1 including truncated region of VC99A01 and VC09N03.
[c]Estimated by Gblocks-processed dataset excluded the tentatively classification sequences (electronic supplementary material, table S1).

## 3.2. Sequences of torque teno virus ORF1

A total of 14 TTV clones from the Taiwanese general population were isolated. There were five clones isolated from sample VC09 and nine clones isolated from sample VC99 for further detailed study. The lengths of putative ORF1 from our sequences and their identity comparison with the closest sequence from GenBank are listed in table 1. TTV ORF1 from our sequences ranging between 735 and 775 amino acids, except VC09N03 and VC99A01 that have premature mutations (table 1). The lowest identity of new sequences compared with known TTVs is 81% (table 1), and the lower boundary of inter-TTV groups p-distance is 0.3750 [9] The 62.5% identity suggested no new TTV group has been found in these 14 sequences. It should be noted that the 75.96% border of intra TTV group 6 identity does not mean group 6 is more conserved or less diverse than other groups (table 1) and be cautioned was expected to of overestimated because current available sequences were very limited. These sequences were deposited in the GenBank and assigned with the accession numbers from MW660794 to MW660807.

## 3.3. Dataset included in the phylogenetic analysis

A total of 193 TTV ORF1 nucleotide sequences from GenBank were aligned with our sequences on the basis of the ORF1 amino acid sequences then processed by Gblocks for phylogenetic analysis. Although Gblocks were set to 'allow all gaps' to accomplish blocks generation, the blocks before the nucleotide position 793 according to TA278 (AB017610) were discarded arbitrarily because there were too many gaps in TA278. For phylogenetic analysis, the concatenated multiplex sequence alignment for tree reconstruction was generated by the following blocks: 793–870, 892–915, 934–1416, 1495–1503, 1675–1695, 1807–2460, 2473–2577 and 2746–2850 nucleotide positions according to prototype TA278.

## 3.4. Phylogenetic analysis

For the concatenated alignment, the GTR + I + $\Gamma$ was determined as the best-fitting model and was applied to ML and Bayesian tree reconstruction. The topology of ML tree is consistent with Bayesian

tree. In our result, seven TTV phylogenetic groups were well supported in both Bayesian and ML trees by posterior probabilities = 1 (figure 1) and bootstrap value greater than 99 (electronic supplementary material, Tree S1), respectively, which is consistent with previous study [9]. The sequences isolated from sample VC09 belonged to group 3b (VC09N02 and VC09N03), group 3c (VC09N93) and group 6 (VC09N27 and VC09N39); sample VC99 were mix-infected with TTV sequences from group 3a (VC99B11), group 4 (VC99B03 and VC99B15) and group 6 (VC99A01, VC99B09, VC99B13, VC99B18, VC99B21 and VC99B24). Both samples represent a mix-infection manner as well as previous studies [9,12].

## 3.5. Genetic characterization

Two sequences from GenBank, P13-1 (KT163896) and P14-1 (KT163900), were closely related to our sequences that gave the opportunity to predict the splicing sites of TTV group 6. The ORF maps of TTV group 6 were similar to P/1C1 (AF298585), which suggested TTV group 6 may have a similar splicing pattern and overlapping genes (figure 2a; electronic supplementary material, figure S1) [38]. Putative splice sites were predicted as 5′ donor site at 628 bp, 2154 bp and 2389 bp as 3′ acceptor sites (figure 2a) by using ASSP, which predicted experimentally confirmed splicing sites of P/1C1 successfully [32]. In addition, the substitutions compared with the representative sequence TW53A26 by another lineage from the same host (TW53A25) and other group 6 sequences from distinct ethnic groups (P13-1, P14-1 and VC09N39) were counted and visualized as a substitution map (figure 2c).

# 4. Discussion

We have first identified a new phylogenetic group of human TTV from eastern Taiwan indigenes [9]. However, considering the world distribution of TTV groups, it seems that TTV group 6 is unlikely to be restricted to Taiwanese indigenes. We developed a TTV group-6-specific PCR for re-screen over TTV positive samples and demonstrated the TTV group 6 also existed in Taiwanese general population. In our result, two group 6 positive samples were randomly picked and cloned. Confirmed by sequencing, both clones were group 6 TTV and thus proved our TTV group-6-specific primers can successfully detect designed PCR target. However, same as the early TTV studies [39], our TTV group-6-specific primers were designed based on the known group 6 TTV sequences, but the diversity of group 6 TTV may beyond our knowledge. Therefore, the true prevalence of group 6 TTV might be underestimated. Since TTV seems abundant in most samples and PCR amplification was easy to perform, and the primers were not designed for quantitative purpose, we did not test the sensitivity of the TTV group-6-specific primers additionally.

Despite the presence of premature mutation, the estimated pairwise dN/dS of VC99A01 and VC09N03 and their closest strain were not higher than those sequences without premature mutation. In addition, the dN/dS of VC99A01 is lower than other VC99 clones, but not closer to 1, suggested the premature mutations did not cause the release of selection pressure and were not fixed yet or came from the part of quasispecies spectrum.

It is interesting to note that two TTV sequences, P13-1 and P14-1, reported by Li *et al.* [40] were classified as group 6 in our result (figure 3). Both P13-1 and P14-1's host were from United States and belonging to distinct ethnic groups [40]. Although P14-1 was isolated from an individual that was obviously different from Taiwanese population, P14-1 was more closely related to TW53A25 than TW53A26, even TW53A25 and TW53A26 were isolated from the same individual (figure 3) [9]. This again proved the TW53A25 and TW53A26 were two lineages that mix-infected in one host. P13-1 and P-14-1 sequences also provide an opportunity to reveal fixed substitutions from closely related sequences.

Although with the exception of the first two arbitrarily excluded blocks, the 5′-blocks (793–870, 892–915 and 934–1416 bp) and 3′-blocks (1807–2460, 2473–2577 and 2746–2850 bp) of Gblocks-processed dataset almost covered the entire analysed regions in the previous study [9]. The blocks 1495–1503 and 1675–1695 were excluded by Hsiao *et al.* by regarding them as hypervariable region (HVR) in previously study [9]; however, the HVR of TTV can be further divided into three segments [41], and both blocks were actually located at the interspace between each HVR segment. HVRs, and both 5′-end and 3′-end ambiguous alignment regions were successfully removed by Gblocks without any need of prior knowledge and the automatic generation of concatenated multiplex sequence alignment can prevent artificial mistake when manual operating of sequences.

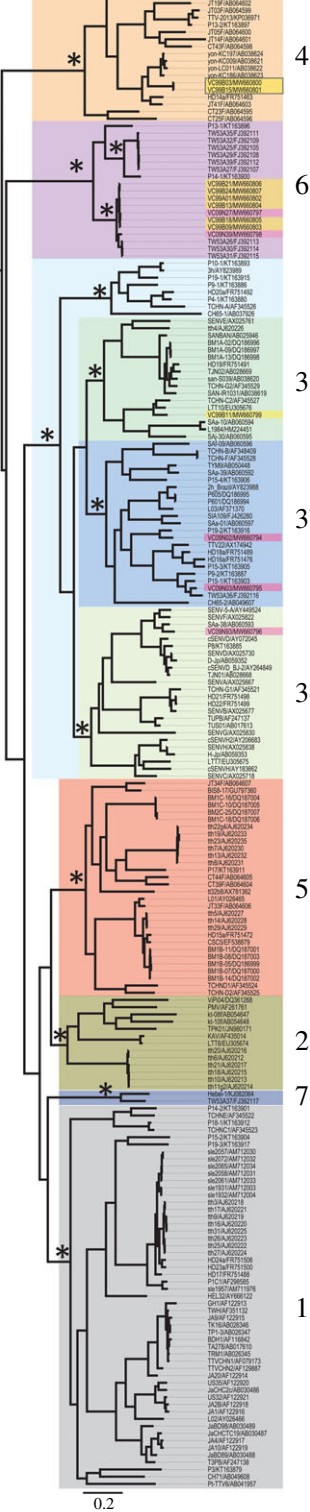

**Figure 1.** Bayesian tree of TTV ORF1 well-aligned region. A Bayesian inferred maximum clade credibility tree that reconstructed on the basis of TTV ORF1 well-aligned region. Grey square indicates group 1. Dark tan square indicates group 2. Light blue indicates group 3 and green, blue and yellow indicate subgroups 3a, 3b and 3c respectively. Orange square indicates group 4. Red square indicates group 5. Purple square indicates group 6. Dark blue indicates group 7. Posterior probabilities are labelled beside stems of each TTV phylogenetic groups. Branches with a * indicate posterior probabilities above 0.95 and are considered as strongly supported. Only major branches were labelled, check electronic supplementary material, Tree S1 for detail. Sequences from this study are labelled by colour bar labels. Pink labels beside taxa names indicate sequences from sample VC09; yellow labels beside taxa names indicate sequences from VC99. The scale bar at the left bottom indicates the evolutionary distance in units of substitutions per site per year.

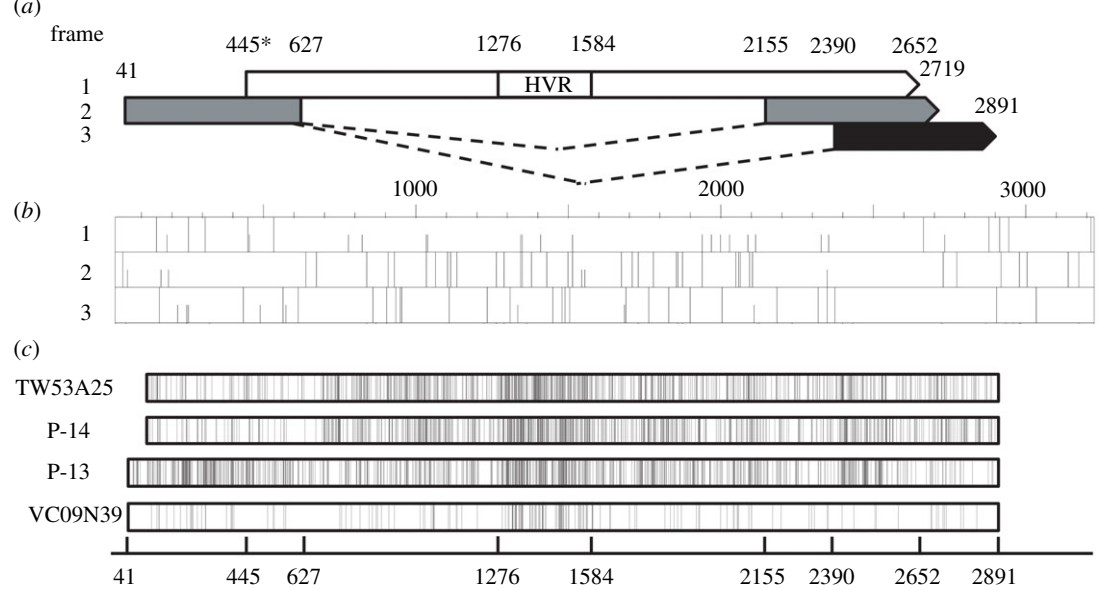

**Figure 2.** The splicing diagram and ORF map of TTV group 6 based on representative sequence TW53A26. Reading frames are indicated at left for both the splicing diagram and ORF map. (*a*) In the splicing diagram, arrow boxes indicate the three distinct reading frames. Putative splicing donor and acceptor sites are linked by dash lines. (*b*) In the ORF map, potential start codons (AUG) are indicated with a half-height line and stop codons (UGA, UAA, UAG) are indicated by full-height lines. (*c*) In the substitutions map, the vertical lines represent nucleotide substitutions compare with representative sequence TW53A26. Sequences that aligned to gaps of TW53A26 were removed and gaps that aligned to TW53A26 were counted as substitutions. * A C → T premature mutation at 523 bp was ignored here.

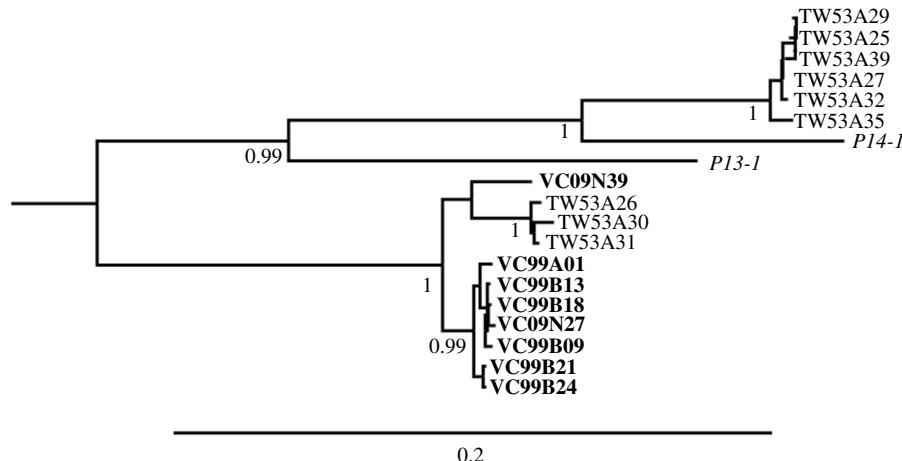

**Figure 3.** A close-up of TTV Group 6 clade of Bayesian tree. Posterior probabilities are labelled beside well-supported branches (greater than 0.95). Sequences from this study are labelled in bold. The scale bar at the bottom indicates evolutionary distance in units of substitutions per site per year. Both sequences from VC09 and VC99 were closely related to TW53A26 clade. In addition, P13-1 and P14-1 (labelled in italic) were submitted by an independent human virome study [40], and also belong to TTV group 6.

The average intra- and inter-group distances were estimated by MEGA5 [33]. The number of base difference per site from averaging over all sequence pairs intra- and inter-groups are shown in table 2. The analysis used Gblocks-processed dataset and excluded the tentative classification sequences (electronic supplementary material, table S1). All positions containing gaps and missing data were eliminated. The groups 6 and 7 have lower intra-group diversity and comparable inter-group distances suggested that current available sequences were very limited and still under-discovered.

We further counted their amino acid substitutions across ORF1 and divided ORF1 into three segments, 5′ to HVR, HVR and HVR to 3′ (table 3). HVR shows higher substitutions to length ratio at amino acid level than non-HVR and has 75% (36 of 48) of total gaps (table 3; electronic supplementary material, Alignment S1 for detail) which was consistent with previous studies [9,41].

**Table 2.** The average intra- and inter-groups distance.

| group | 1 | 2 | 3a | 3b | 3c | 4 | 5 | 6 |
|---|---|---|---|---|---|---|---|---|
| 1 | 0.250 | | | | | | | |
| 2 | 0.284 | 0.418 | | | | | | |
| 3a | 0.245 | 0.462 | 0.457 | | | | | |
| 3b | 0.298 | 0.459 | 0.453 | 0.378 | | | | |
| 3c | 0.285 | 0.451 | 0.444 | 0.394 | 0.388 | | | |
| 4 | 0.272 | 0.475 | 0.479 | 0.497 | 0.487 | 0.488 | | |
| 5 | 0.264 | 0.422 | 0.398 | 0.459 | 0.454 | 0.458 | 0.467 | |
| 6 | 0.140 | 0.464 | 0.451 | 0.477 | 0.473 | 0.472 | 0.488 | 0.457 | |
| 7 | 0.191 | 0.413 | 0.414 | 0.461 | 0.456 | 0.454 | 0.473 | 0.409 | 0.464 |

**Table 3.** Pairwise amino acid (a.a.) substitution counts of TTV ORF1.

| sequence | TW53A25 | P14-1 | P13-1 | VC99A01 | TW53A26 | length (a.a) |
|---|---|---|---|---|---|---|
| (A) | | | 5′ to HVR | | | |
| TW53A25 | ID | 27 | 69 (5) | 71 | 71 | 277 |
| P14-1 | 27 | ID | 67 (5) | 66 | 68 | 277 |
| P13-1 | 69 (5) | 67 (5) | ID | 62 (5) | 61 (5) | 276 |
| VC09N39 | 71 | 66 | 62 (5) | ID | 11 | 277 |
| TW53A26 | 71 | 68 | 61 (5) | 11 | ID | 277 |
| (B) | | | HVR | | | |
| TW53A25 | ID | 39 | 79 (18) | 97 (36) | 105 (36) | 139 |
| P14-1 | 39 | ID | 78 (18) | 94 | 101 | 139 |
| P13-1 | 79 (18) | 78 (18) | ID | 76 (18) | 79 (18) | 121 |
| VC09N39 | 97 (36) | 94 | 76 (18) | ID | 30 | 103 |
| TW53A26 | 105 (36) | 101 | 79 (18) | 30 | ID | 103 |
| (C) | | | HVR to 3′ | | | |
| TW53A25 | ID | 36 | 93 (6) | 91 (2) | 91 (2) | 354 |
| P14-1 | 36 | ID | 93 (6) | 95 (2) | 94 (2) | 354 |
| P13-1 | 93 (6) | 93 (6) | ID | 101 (6) | 98 (6) | 350 |
| VC09N39 | 91 (2) | 95 (2) | 101 (6) | ID | 11 | 356 |
| TW53A26 | 91 (2) | 94 (2) | 98 (6) | 11 | ID | 356 |

5′ to HVR region (A), HVR region (B) and HVR to 3′ region (C) of TTV group 6 representative sequences from Taiwan and Unite States. Numbers in brackets indicated how many gaps (indels) were including in substitution counts. The HVR region was operationally defined as 278–380 a.a. according to the ORF1 of TW53A26.

The substitution map also showed the most density in the HVR (figure 2c). However, the substitutions beside HVR were not evenly distributed (figure 2c).

It has been reported that TTV encodes overlapping proteins [38]. The 5′ coding region of ORF1 is overlapped with a frame 2 coding region and the 3′ coding region of ORF1 is partially overlapped with a frame 2 coding region and partially triple overlapped with both frame 2 and frame 3 coding region (figure 2a). Overlapping regions tend to lower diversity than non-overlapping regions because of hypersensitivity to deleterious mutation [42]. However, the triple overlapped region showed more substitutions density next to HVR (figure 2c). We further present this phenomenon by using the ORF1 nucleotide sequences alignment of all TTV group 6 clones. The average diversity of each region was taken by calculating DNA pairwise identity matrix of TTV group 6 to average. The ORF1, excluding

HVR, were divided into five regions: 5′ overlapping region (445–627 bp), 5′ non-overlapping region (628–1275 bp), 3′ non-overlapping region (1585–2154 bp), 3′ overlapping region (2155–2389 bp) and 3′ triple overlapping region (2390–2652 bp), according to representative sequence TW53A26 (see electronic supplementary material, Alignment S2 for detail). The average diversity of non-overlapping regions (15.61% and 14.98% for 5′ and 3′ overlapping region, respectively) were higher than overlapping regions (7.53% and 9.78% for 5′ and 3′ overlapping region, respectively). By contrast, the average diversity of the triple overlapping region (13.46%) was slightly lower then the non-overlapping region. Theoretically, any mutation occurred at a triple overlapping gene will cause amino acid substations at one or more frames. This suggested a triple overlapping region will be even more conserved than overlapping region, such as bovine leukaemia virus's pXBL region [43]. The highly diverse triple overlapping region may represent a unique phenomenon of TTV.

This study implied that TTV group 6 may be prevalent worldwide regardless of the geographic region and/or ethnic groups. The intra- and inter-TTV groups mix-infection were not only represented in Taiwan but also in many countries. The length variation of ORF1 along with TTV divergence suggested that coding region indels were one of the major driving forces of TTV evolution and result in TTV's diversity.

Ethics. The study protocol was approved by the Institutional Review Board of MacKay Memorial Hospital (MMH-I-S-460).

Data accessibility. The datasets used and analysed during the current study are available from the corresponding author on reasonable request. DNA sequences: Genbank accessions MW660794 to MW660807 (this study). Phylogenetic data were included in electronic supplementary material, Tree S1.

The data are provided in the electronic supplementary material [44].

Authors' contributions. H.-F.L. was involved in conceptualization, resources, formal analysis, writing the review and editing, supervision and funding. K.-L.H. was involved in conceptualization, methodology, investigation, validation, formal analysis, data curation, writing the original draft preparation and visualization. Y.-J.C. and C.-L.L. were involved in investigation and validation. L.-Y.W. and J.-C.C. were involved in writing the review and editing.

Competing interests. The authors declare no conflicts of interest.

Funding. This study was supported by the Ministry of Science and Technology, Taiwan (grant no. NSC 96-2314-B-195-010) and in part by the MacKay Memorial Hospital, Taiwan (grant nos. MMH-107-24 and 2019-C01-00055).

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
