## [Peer Review File · Royal Society Open Science]

Review History

RSOS-210938.R0 (Original submission)

Review form: Reviewer 1

Is the manuscript scientifically sound in its present form?

Yes

Are the interpretations and conclusions justified by the results?

Yes

Is the language acceptable?

Yes

Do you have any ethical concerns with this paper?

No

Have you any concerns about statistical analyses in this paper?

No

Recommendation?

Accept with minor revision (please list in comments)

Comments to the Author(s)

The manuscript by Hsiao et al. is of some interest in providing new data on TTV, which is the focus of several studies for its potential role as an immune system' biomarker in several settings of patients. Overall, the data are scientifically sound and concisely and clearly described.

However, there are points that need to be addressed.

The first point is the taxonomy of Anellovirus and TTV. The authors used a not updated classification of the virus, thus I suggest describing the data considering this revised taxonomy (by ICTV). The second point is the need to have more information about the sensitivity and specificity of the PCRs described in the paper.

Review form: Reviewer 2**Is the manuscript scientifically sound in its present form?**

No

Are the interpretations and conclusions justified by the results?

No

Is the language acceptable?

Yes

Do you have any ethical concerns with this paper?

No

Have you any concerns about statistical analyses in this paper?

No

Recommendation?

Major revision is needed (please make suggestions in comments)

Comments to the Author(s)

The present study aimed at the development specific PCR for detection TTV group 6 in Taiwanese population and the genetic characterization of their sequences.

This study is interesting, however, from my point of view, it is necessary to attend some corrections and suggestions.

In the abstract section

Line 29: "Torque teno virus (TTV) is the most common human virus" This statement is very risky; I suggest reconsidering it or in the introduction provide information to support it

In the introduction section

Line 55: Use the ICTV taxonomic rules

In the methods section

In L-89. Explain how they got only the viral DNA?

In L-90-91. Explain if it was a modified protocol or following the manufacturer's instructions.

In L-99. Describe the PCR reaction mix and the concentration of DNA used and indicate the PCR positive controls used

In L-100. Change "using the program" to "amplification conditions."

In L-103. Include the expected amplicon size.

In L-127. Describe in material and methods how the isolation of the 14 viruses included in table 1 was carried out.

To complete the genetic characterization, it is important to include a genetic distance analysis and determine synonymous and nonsynonymous substitutions (dN/dS).

In the results section

In L-150-152. This is not clear, the methods section describes the use of a PCR to amplify the untranslated region of TTV group 6. On the other hand, the results section describes the detection of TTV in more than 86% of patients samples studied and only 4.6% of TTV from group 6. This implies the use of more than one PCR that must be properly described in the methods section.

In L-158-160. There is no description in the methods section of the isolation of the different strains that are described in results. Include this information. If the sequences obtained in the study to be referred to as isolated, this definition is incorrect

In L-167. The title of Table 1 is not very descriptive and the information is very poor and confusing. The genetic distance between all TTV groups and subgroups should be included. In addition, a dN / dS analysis should be performed between the sequences obtained in the study.

In L-186-192. What strategy was used to ensure that the viral genetic diversity of the group of samples analyzed is represented?, for example, limiting dilution PCR.

In L-194. Include the access number in the sentence "Two strains from GenBank (P13-1 and P14-1) were distinct"

In L-203. Revise the next sentence "distinct ethnic groups"

In the discussion section

In L-212-214. The idea that is intended to be discussed in these lines is not clear, revise and rewrite. A discussion including genetic distance and analysis of synonymous and nonsynonymous mutations is suggested.

In L-231, 232, 233, 234, 239, 240, 243. What does the word HVR mean?

In L-270. Table 2 should be elaborated again considering the analysis by genetic group. This information would allow us to understand the genetic diversity among TTVs, and the diversity among group 6 of TTVs.

In L-287. Figure legends should be placed after the references.

The phylogenetic tree should be built again with the accession number and name of the sequences, including the study sequences with their respective accession number. A phylogram

with 100 sequences is possible to clearly see the group and sequences that make up each clade. The trees included in this study do not allow the analysis and visualization of the association of the sequences between each branch of the tree.

Decision letter (RSOS-210938.R0)

Dear Dr Hsiao,

The Editors assigned to your paper RSOS-210938 "Detection and genetic characterization of the novel torque teno virus group 6 in Taiwanese general population" have now received comments from reviewers and would like you to revise the paper in accordance with the reviewer comments and any comments from the Editors. Please note this decision does not guarantee eventual acceptance.

Both reviewers are positive about the paper, but each raises some substantive comments that will require careful consideration. We invite you to respond to the comments supplied below and revise your manuscript. Below the referees' and Editors' comments (where applicable) we provide additional requirements. Final acceptance of your manuscript is dependent on these requirements being met. We provide guidance below to help you prepare your revision.

Please submit your revised manuscript and required files (see below) no later than 21 days from today's (ie 26-Jul-2021) date. Note: the ScholarOne system will 'lock' if submission of the revision is attempted 21 or more days after the deadline. If you do not think you will be able to meet this deadline please contact the editorial office immediately.

on behalf of Dr Cheryl Andam (Associate Editor) and Steve Brown (Subject Editor)
openscience@royalsociety.org

Reviewer comments to Author:

Reviewer: 1

Comments to the Author(s)

The manuscript by Hsiao et al. is of some interest in providing new data on TTV, which is the focus of several studies for its potential role as an immune system' biomarker in several settings of patients. Overall, the data are scientifically sound and concisely and clearly described.

However, there are points that need to be addressed.

The first point is the taxonomy of Anellovirus and TTV. The authors used a not updated classification of the virus, thus I suggest describing the data considering this revised taxonomy (by ICTV). The second point is the need to have more information about the sensitivity and specificity of the PCRs described in the paper.

Reviewer: 2

Comments to the Author(s)

The present study aimed at the development specific PCR for detection TTV group 6 in Taiwanese population and the genetic characterization of their sequences.

This study is interesting, however, from my point of view, it is necessary to attend some corrections and suggestions.

In the abstract section

Line 29: "Torque teno virus (TTV) is the most common human virus" This statement is very risky; I suggest reconsidering it or in the introduction provide information to support it

In the introduction section

Line 55: Use the ICTV taxonomic rules

In the methods section

In L-89. Explain how they got only the viral DNA?

In L-90-91. Explain if it was a modified protocol or following the manufacturer's instructions.

In L-99. Describe the PCR reaction mix and the concentration of DNA used and indicate the PCR positive controls used

In L-100. Change "using the program" to "amplification conditions.

In L-103. Include the expected amplicon size.

In L-127. Describe in material and methods how the isolation of the 14 viruses included in table 1 was carried out.

To complete the genetic characterization, it is important to include a genetic distance analysis and determine synonymous and nonsynonymous substitutions (dN/dS).

In the results section

In L-150-152. This is not clear, the methods section describes the use of a PCR to amplify the untranslated region of TTV group 6. On the other hand, the results section describes the detection of TTV in more than 86% of patients samples studied and only 4.6% of TTV from group 6. This implies the use of more than one PCR that must be properly described in the methods section.

In L-158-160. There is no description in the methods section of the isolation of the different strains that are described in results. Include this information. If the sequences obtained in the study to be referred to as isolated, this definition is incorrect

In L-167. The title of Table 1 is not very descriptive and the information is very poor and confusing. The genetic distance between all TTV groups and subgroups should be included. In addition, a dN / dS analysis should be performed between the sequences obtained in the study.

In L-186-192. What strategy was used to ensure that the viral genetic diversity of the group of samples analyzed is represented?, for example, limiting dilution PCR.

In L-194. Include the access number in the sentence "Two strains from GenBank (P13-1 and P14-1) were distinct"

In L-203. Revise the next sentence "distinct ethnic groups"

In the discussion section

In L-212-214. The idea that is intended to be discussed in these lines is not clear, revise and rewrite. A discussion including genetic distance and analysis of synonymous and nonsynonymous mutations is suggested.

In L-231, 232, 233, 234, 239, 240, 243. What does the word HVR mean?

In L-270. Table 2 should be elaborated again considering the analysis by genetic group. This information would allow us to understand the genetic diversity among TTVs, and the diversity among group 6 of TTVs.

In L-287. Figure legends should be placed after the references.

The phylogenetic tree should be built again with the accession number and name of the sequences, including the study sequences with their respective accession number. A phylogram with 100 sequences is possible to clearly see the group and sequences that make up each clade. The trees included in this study do not allow the analysis and visualization of the association of the sequences between each branch of the tree.

===PREPARING YOUR MANUSCRIPT===

Your revised paper should include the changes requested by the referees and Editors of your manuscript. You should provide two versions of this manuscript and both versions must be provided in an editable format:
 one version identifying all the changes that have been made (for instance, in coloured highlight, in bold text, or tracked changes);
 a 'clean' version of the new manuscript that incorporates the changes made, but does not highlight them. This version will be used for typesetting if your manuscript is accepted.

Please ensure that you include an acknowledgements' section before your reference list/bibliography. This should acknowledge anyone who assisted with your work, but does not

qualify as an author per the guidelines at <https://royalsociety.org/journals/ethics-policies/openness/>.

===PREPARING YOUR REVISION IN SCHOLARONE===

-- Ensure that your data access statement meets the requirements at <https://royalsociety.org/journals/authors/author-guidelines/#data>. You should ensure that you cite the dataset in your reference list. If you have deposited data etc in the Dryad repository, please include both the 'For publication' link and 'For review' link at this stage.

Author's Response to Decision Letter for (RSOS-210938.R0)

See Appendix A.

RSOS-210938.R1 (Revision)

Review form: Reviewer 1

Is the manuscript scientifically sound in its present form?

Yes

Are the interpretations and conclusions justified by the results?

Yes

Is the language acceptable?

Yes

Do you have any ethical concerns with this paper?

No

Have you any concerns about statistical analyses in this paper?

No

Recommendation?

Accept as is

Comments to the Author(s)

All points raised have been well stressed by the authors

Review form: Reviewer 2**Is the manuscript scientifically sound in its present form?**

Yes

Are the interpretations and conclusions justified by the results?

Yes

Is the language acceptable?

Yes

Do you have any ethical concerns with this paper?

No

Have you any concerns about statistical analyses in this paper?

No

Recommendation?

Accept as is

Comments to the Author(s)

Thanks to the authors for considering the recommendations and incorporating them into their manuscript.

Decision letter (RSOS-210938.R1)

Dear Dr Hsiao,

It is a pleasure to accept your manuscript entitled "Detection and genetic characterization of the novel torque teno virus group 6 in Taiwanese general population" in its current form for publication in Royal Society Open Science.

on behalf of Dr Cheryl Andam (Associate Editor) and Steve Brown (Subject Editor)
openscience@royalsociety.org

Reviewer comments to Author:

Reviewer: 1

Comments to the Author(s)

All points raised have been well stressed by the authors

Reviewer: 2

Comments to the Author(s)

Thanks to the authors for considering the recommendations and incorporating them into their manuscript.

Appendix A

Reviewer comments to Author:

Reviewer: 1

Comments to the Author(s)

Reviewer comments:

The manuscript by Hsiao et al. is of some interest in providing new data on TTV, which is the focus of several studies for its potential role as an immune system' biomarker in several settings of patients. Overall, the data are scientifically sound and concisely and clearly described. However, there are points that need to be addressed.

Q: The first point is the taxonomy of Anellovirus and TTV. The authors used a not updated classification of the virus, thus I suggest describing the data considering this revised taxonomy (by ICTV).

A:

We have replaced anelloviruses family by family *Anelloviridae*. However, the classification of phylogenetic groups is a system between species and genera of ICTV classification. So far, the “species” of ICTV classification still cannot cover all of reported strains. It is therefore very difficult to describe the classification according to ICTV system solely, thus we cannot modify much in the manuscript.

Q: The second point is the need to have more information about the sensitivity and specificity of the PCRs described in the paper.

A:

We have added a few sentences concerning about the sensitivity and specificity of the PCR in the manuscript. Our TTV group-6-specific primers were designed based on the known group 6 TTV sequences in an alignment that containing hundreds of different TTV sequences. We believe the primers' locations have enough specificity for group 6 TTV only. Because we did not have samples containing all the known genotypes, so we cannot test the specificity in real. Since TTV seems abundant in most samples and PCR amplification was easy to perform, and the design of the primers was not for quantitative purpose, we did not test the sensitivity of the TTV group-6-specific primers additionally.

Reviewer: 2

Comments to the Author(s)

Reviewer comments:

The present study aimed at the development specific PCR for detection TTV group 6 in Taiwanese population and the genetic characterization of their sequences.

This study is interesting, however, from my point of view, it is necessary to attend some corrections and suggestions.

In the abstract section

Q:

Line 29: “Torque teno virus (TTV) is the most common human virus” This statement is very risky; I suggest reconsidering it or in the introduction provide information to support it

A:

We have added “one of” the most common human virus in the Abstract and Introduction section.

Q:

Line 55: Use the ICTV taxonomic rules

A:

We replaced anelloviruses family by family *Anelloviridae*.

In the methods section

Q:

In L-89. Explain how they got only the viral DNA?

A:

The kit will extract all nucleic acid when using whole blood samples. We got (almost) only the viral DNA because we using serum samples.

Q:

In L-90-91. Explain if it was a modified protocol or following the manufacturer's instructions.

A:

Because the amount of serum samples was very limited, we used half amount of serum and adjusted the volume to 200 μ l with Phosphate-buffered saline to fit the manufacturer's instructions. We added the description in the methods section.

Q:

In L-99. Describe the PCR reaction mix and the concentration of DNA used and indicate the PCR positive controls used

A:

We have added description of PCR reaction mix and positive control for each PCR reaction.

Q:

In L-100. Change "using the program" to "amplification conditions."

A: We have changed.

Q:

In L-103. Include the expected amplicon size.

A:

We have added the location of amplicon on group 6 TTV to show its size and primers' location.

Q:

In L-127. Describe in material and methods how the isolation of the 14 viruses included in table 1 was carried out.

A: We picked 2 of TTV group 6 positive samples randomly and have described at "Amplification and sequencing of TTV ORF1" section of Methods. Since TTVs were mix-infected in every single person and no universal sequencing primer can be designed for ORF1 region, every single clone has to be sequenced independently by

primer walking strategy. It was highly time consuming and expensive. Therefore we only performed ORF1 sequencing on 14 clones isolated from 2 of group-6-positive PCR products as description in the Results.

Reviewer comments:

To complete the genetic characterization, it is important to include a genetic distance analysis and determine synonymous and nonsynonymous substitutions (dN/dS).

In the results section

Q:

In L-150-152. This is not clear, the methods section describes the use of a PCR to amplify the untranslated region of TTV group 6. On the other hand, the results section describes the detection of TTV in more than 86% of patients samples studied and only 4.6% of TTV from group 6. This implies the use of more than one PCR that must be properly described in the methods section.

A:

We apologize for the confusion. Actually there were 3 different PCRs: detection for TTV positive, detection for TTV group 6 from TTV positive samples, and amplification and cloning of TTV-group 6 ORF1 sequences from TTV-group 6 positive samples. We have rewritten into different paragraphs in the Method section to clarify this issue.

Q:

In L-158-160. There is no description in the methods section of the isolation of the different strains that are described in results. Include this information. If the sequences obtained in the study to be referred to as isolated, this definition is incorrect

A:

We apologize for the confusion. We have changed the word to sequences and clones.

Q:

In L-167. The title of Table 1 is not very descriptive and the information is very poor and confusing. The genetic distance between all TTV groups and subgroups should be included. In addition, a dN / dS analysis should be performed between the sequences obtained in the study.

A:

We have performed dN / dS analysis and provided intra group identity to the table. The title of Table 1 has been changed to “Genetic diversity and dN/dS ratio of the TTV clone sequences and their closest sequence from GenBank database”.

Q:

In L-186-192. What strategy was used to ensure that the viral genetic diversity of the group of samples analyzed is represented?, for example, limiting dilution PCR.

A:

We did not use limiting dilution PCR for quantization. Instead we just cloned all what we can clone and put all of the cloned sequences into this study. We have added a notice at discussion to state that the study was not designed for quantitative purpose.

Q:

In L-194. Include the access number in the sentence “Two strains from GenBank (P13-1 and P14-1) were distinct”

A: We have added the access numbers.

Q:

In L-203. Revise the next sentence “distinct ethnic groups”

A:

Thank you for point out the typos. We have changed to ethnic groups.

Q:

In L-212-214. The idea that is intended to be discussed in these lines is not clear, revise and rewrite. A discussion including genetic distance and analysis of synonymous and nonsynonymous mutations is suggested.

A:

We have rewritten this part according to the addition result of dN / dS analysis.

Q:

In L-231, 232, 233, 234, 239, 240, 243. What does the word HVR mean?

A: We have added full word when it first appeared.

Q:

In L-270. Table 2 should be elaborated again considering the analysis by genetic group. This information would allow us to understand the genetic diversity among TTVs, and the diversity among group 6 of TTVs.

A: For easy to understand the genetic diversity among TTVs, we have added a new Table 2 to present the average diversity intra and inter all TTV groups. Table 3 (originally Table 2) was intended to present diversity within group 6 TTVs. When group 6 was compared to, e.g. group 1, there will be hundreds of substitutions and is not possible to be shown clearly. Therefore we added a new Table 2 for instead.

Q:

In L-287. Figure legends should be placed after the references.

A: We have moved.

Q:

The phylogenetic tree should be built again with the accession number and name of the sequences, including the study sequences with their respective accession number. A phylogram with 100 sequences is possible to clearly see the group and sequences that make up each clade. The trees included in this study do not allow the analysis and visualization of the association of the sequences between each branch of the tree.

A: We re-reconstructed a phylogram in rectangle form with sequences' accession number and upload the tree file as supplementary file to allow further analysis.

===PREPARING YOUR MANUSCRIPT===

Your revised paper should include the changes requested by the referees and Editors

of your manuscript. You should provide two versions of this manuscript and both versions must be provided in an editable format:

- one version identifying all the changes that have been made (for instance, in coloured highlight, in bold text, or tracked changes);
- a 'clean' version of the new manuscript that incorporates the changes made, but does not highlight them. This version will be used for typesetting if your manuscript is accepted.

===PREPARING YOUR REVISION IN SCHOLARONE===

-- Ensure that your data access statement meets the requirements

at <https://royalsociety.org/journals/authors/author-guidelines/#data>. You should

ensure that you cite the dataset in your reference list. If you have deposited data etc in the Dryad repository, please include both the 'For publication' link and 'For review'

link at this stage.

-- If you have uploaded ESM files, please ensure you follow the guidance at <https://royalsociety.org/journals/authors/author-guidelines/#supplementary-material> to include a suitable title and informative caption. An example of appropriate titling and captioning may be found

at https://figshare.com/articles/Table_S2_from_Is_there_a_trade-off_between_peak_performance_and_performance_breadth_across_temperatures_for_aerobic_scope_in_tropical_fishes_/3843624.
